# Pharmacometabolomics in TB meningitis— Understanding the pharmacokinetic, metabolic, and immune factors associated with anti-TB drug concentrations in cerebrospinal fluid

Jeffrey M. Collins[1]*, Maia Kipiani[2,3,4], Yutong Jin[5], Ashish A. Sharma[6], Jeffrey A. Tomalka[6], Teona Avaliani[2], Mariam Gujabidze[2], Tinatin Bakuradze[2], Shorena Sabanadze[2], Zaza Avaliani[2,7], Henry M. Blumberg[1,8], David Benkeser[5], Dean P. Jones[1], Charles Peloquin[9], Russell R. Kempker[1]

1 Department of Medicine, Emory University School of Medicine, Atlanta, Georgia, United States of America, 2 National Center for Tuberculosis and Lung Diseases, Tbilisi, Georgia, 3 The University of Georgia, Tbilisi, Georgia, 4 David Tvildiani Medical University, The University of Georgia, Tbilisi, Georgia, 5 Department of Biostatistics, Rollins School of Public Health of Emory University, Atlanta, Georgia, United States of America, 6 Department of Pathology and Laboratory Medicine, Emory University School of Medicine, Atlanta, Georgia, United States of America, 7 European University, Tbilisi, Georgia, 8 Departments of Epidemiology and Global Health, Rollins School of Public Health, Emory University, Atlanta, Georgia, United States of America, 9 Department of Pharmacotherapy and Translational Research, College of Pharmacy, University of Florida, Gainesville, Florida, United States of America

* jeffrey.michael.collins@emory.edu

## Abstract

Poor penetration of many anti-tuberculosis (TB) antibiotics into the central nervous system (CNS) is thought to be a major driver of morbidity and mortality in TB meningitis (TBM). While the amount of a particular drug that crosses into the cerebrospinal fluid (CSF) varies from person to person, little is known about the host factors associated with interindividual differences in CSF concentrations of anti-TB drugs. In patients diagnosed with TBM from the country of Georgia (n = 17), we investigate the association between CSF concentrations of anti-TB antibiotics and multiple host factors including serum drug concentrations and CSF concentrations of metabolites and cytokines. We found >2-fold differences in CSF concentrations of anti-TB antibiotics from person to person for all drugs tested including cycloserine, ethambutol, imipenem, isoniazid, levofloxacin, linezolid, moxifloxacin, pyrazinamide, and rifampin. While serum drug concentrations explained over 30% of the variation in CSF drug concentrations for cycloserine, isoniazid, linezolid, and pyrazinamide (adjusted $R^2 \geq 0.3$, $p < 0.001$ for all), there was no significant association between serum concentrations of imipenem and ethambutol and their respective CSF concentrations. CSF concentrations of carnitines were significantly associated with concentrations of ethambutol and imipenem ($q < 0.05$), and imipenem was the only antibiotic significantly associated with CSF cytokine concentrations. These results indicate that there is high interindividual variability in CSF drug concentrations in patients treated for TBM, which is only partially explained by differences in serum drug concentrations. With the exception

**Data availability statement:** All relevant data are within the manuscript and its Supporting Information files.

**Funding:** This work was supported by grants from the National Institutes of Health (NIH) and National Institute of Allergy and Infectious Diseases [R03AI139871, K23AI103044, K23AI144040, P30AI168386, P30AI050409]; NIH Fogarty International Center [D43TW007124]; and NIH National Center for Advancing Translational Science [UL1TR002378], Bethesda, MD, USA. The funders had no role in study design, data collection and analysis, decision to publish, or preparation of the manuscript.

**Competing interests:** The authors have declared that no competing interests exist.

of imipenem, there was no association between CSF drug concentrations and concentrations of cytokines and chemokines.

## Introduction

Tuberculosis meningitis (TBM) is the most lethal manifestation of tuberculosis (TB) disease, with a mortality rate $\geq 25\%$ in those with drug susceptible disease and $> 65\%$ in persons with drug resistance [1,2]. Therapeutic options for TBM are more limited than in other forms of TB due to poor penetration of many anti-TB drugs into the central nervous system (CNS) [1,3–5]. However, the concentration of antibiotics in the CNS can vary widely by antibiotic and from person to person, indicating certain endotypes exist that lead to favorable CNS penetration of drugs. While it is generally thought that meningeal inflammation increases the CNS concentration of antibiotics, there is limited empirical evidence supporting this concept in TBM [6]. Further, little is known about the specific inflammatory signaling networks in TBM that may modulate CNS penetration of anti-TB chemotherapeutic agents. An improved understanding of host responses associated with CNS penetration of anti-TB drugs could help inform new strategies to enhance drug delivery to the site of disease in TBM.

Recent advances in metabolomics and high-density cytokine measurement allow for high-resolution immunometabolic profiling of a variety of human samples including cerebrospinal fluid (CSF) [7]. Such profiling allows for simultaneous measurement of over 30 cytokines and thousands of metabolites, including xenobiotics [8] and molecules that regulate inflammation [9,10]. Combining measurement of these host response molecules with drug concentrations at the site of disease has the potential to elucidate metabolic and inflammatory processes that regulate drug concentrations and efficacy in TBM; an approach termed pharmacometabolomics [11].

While the average proportion of many first-line anti-TB drugs that reach the CSF from blood is known [12,13], CSF drug concentrations are also known to vary widely from person to person. More work is needed to better understand the factors that produce such variability in order to enhance CNS drug delivery during TBM treatment. In a well characterized population of patients from the country of Georgia diagnosed and treated for TBM, we integrated data on serum and CSF concentrations of anti-TB antibiotics [5] and CSF concentrations of endogenous metabolites and cytokines [7]. By integrating these data sets, we sought to address multiple knowledge gaps that currently exist with regard to antibiotic therapy for TBM as follows: 1) the degree of interindividual variability in CSF drug concentrations, 2) whether variation in serum drug concentrations explain the variation in CSF drug concentrations, 3) whether untargeted metabolomics can be used to accurately measure concentrations of TB antibiotics in the CNS, and 4) whether individual differences in the CSF milieu of soluble immunometabolic mediators (i.e., metabolites and cytokines) are associated with CSF drug concentrations of different anti-TB drugs.

## Materials and methods

### Setting and participants with TBM

Persons with TBM were enrolled from the National Center for Tuberculosis and Lung Diseases (NCTLD) in Tbilisi, Georgia as part of a clinical pharmacology study evaluating the penetration of anti-TB drugs into the CNS [5]. Patients aged $\geq 16$ years treated in the NCTLD adult TBM ward from June 12, 2018 to December 31, 2019 were eligible for inclusion. All

patients suspected of having TBM underwent a lumbar puncture; acid-fast bacilli (AFB) staining, liquid and solid culture, and Xpert MTB/RIF assay were performed on CSF. As per standard of care for patients hospitalized with TBM at NCTLD, lumbar punctures were performed at approximately 7, 14, and 28 days after TBM treatment initiation and monthly for as long as patients were hospitalized to follow CSF cell and protein counts in response to treatment [5,14]. Of the 72 CSF samples analyzed, 63 (88%) were obtained within the first 2 months of treatment. Treatment regimens were selected by treating clinicians based on treatment history, comorbidities, and drug susceptibility results when available [14]. All patients also received a 6–8-week course of dexamethasone (400-1200 mg). Written informed consent was obtained from all study participants, and study approval was obtained from the institutional review boards of Emory University and the NCTLD.

## Sample collection and drug quantification

Blood samples were collected at 2 and 6 hours after drug administration at each timepoint. For CSF collection, 3 mL was collected at each time point, alternating between 2 and 6 hours after drug administration to capture early and delayed drug penetration into the CSF. All centrifuged blood as well as CSF samples were stored at –80°C at the NCTLD until shipped to the Infectious Diseases Pharmacokinetic Laboratory at the University of Florida, where drug concentrations were quantified. Total concentrations of each drug were measured using validated liquid chromatography tandem mass spectrometry assays. The assays were validated for human plasma, and cross checked for matrix effects using artificial CSF. The analyses were performed on Thermo Scientific TSQ Endura or TSQ Quantum Ultra mass spectrometers.

## Metabolomics analysis

De-identified CSF samples were randomized by a computer-generated list into blocks of 40 samples prior to transfer to the analytical laboratory where personnel were blinded to clinical and demographic data. Thawed CSF (65 μL) was treated with 130 μl acetonitrile (2:1, v/v) containing an internal isotopic standard mixture (3.5 μL/sample), as previously described [15]. Samples were centrifuged and supernatants were analyzed using an Orbitrap Q Exactive Mass Spectrometer (Thermo Scientific, San Jose, CA, USA) with dual HILIC positive and c18 negative liquid chromatography (Higgins Analytical, Targa, Mountain View, CA, USA, 2.1 x 10 cm) with a formic acid/acetonitrile gradient. The high-resolution mass spectrometer was operated over a scan range of 85 to 1275 mass/charge ($m/z$) [16]. Data were extracted and aligned using apLCMS [17] and xMSanalyzer [18] with each feature defined by specific $m/z$ value, retention time, and integrated ion intensity [16]. Three technical replicates were performed for each CSF sample and intensity values were median summarized. Peaks for anti-TB antibiotics in the untargeted metabolomics data were identified using accurate mass and retention time measurements. Peaks were observed for all antibiotics except cycloserine.

## Cytokine detection

The commercially available U-PLEX assay by Meso Scale Discovery (MSD) was used for CSF cytokine detection. This assay allows for the evaluation of multiplexed biomarkers by using custom made U-PLEX sandwich antibodies with a SULFO-TAG conjugated antibody and electrochemiluminescence (ECL) detection. Direct quantitation of cytokines was performed using standard curves generated by 4-fold serial dilutions of standard calibrators provided by MSD. Plates were read on the QuickPlex SQ 120 using Methodical Mind™ software and plate data analyzed using Discovery Workbench™ software (MSD) [19,20].

## Statistical analysis

Descriptive statistics were provided for baseline characteristics. For each drug, we determined the clinical variables associated with serum and CSF drug concentrations by applying a linear mixed regression model using the R package nlme [21]. Covariates examined included age, sex, weight, creatinine, sampling time (2 or 6 hours), and serum drug concentrations. For time points where weight or creatinine values were missing, values were imputed using the R package mitml, which provides tools for multiple imputation of missing data in multilevel modeling [22]. For each antibiotic concentration in the CSF, we reported the adjusted $R^2$ values of the pairwise comparison of the univariate models, whether they included or excluded serum concentration as a covariate [23]. Similarly, to determine associations between CSF drug concentrations and metabolite and cytokine concentrations, we applied a linear mixed regression model, adjusting for sampling time (2 or 6 hours), and reported the adjusted $R^2$ values of each pairwise comparison of models, whether they included or excluded the concentration of a particular metabolite or cytokine [23]. A false discovery rate (FDR) correction was employed to account for multiple comparisons when examining associations between anti-TB drugs and metabolites and cytokines [20]. All analyses were conducted using R version 4.2.1. All datasets used for this analysis are available in the online supplement (S1 File).

## Results

### Participants

We studied CSF samples from 17 participants with suspected TBM presenting to the National Center for Tuberculosis and Lung Diseases (NCTLD) (Table 1). Among the 17 patients with suspected TBM, five (29%) were microbiologically confirmed while three (18%) were considered to have probable TBM and nine (53%) were considered to have possible TBM based on clinical and laboratory criteria [24,25]. The median CSF white blood cell (WBC) count at diagnosis was 209 cells/mm³ (IQR 130-286), with 94% lymphocytes (IQR 85-96), a median glucose of 40 mg/dL (IQR 21-50), and total protein of 99 mg/dL (IQR 66-132). Antibiotic regimens were selected by treating clinicians based on NCTLD guidelines and response to treatment as previously reported [5,14]. Five persons with TBM were deemed to have a poor initial response to treatment, two of whom were diagnosed with multi-drug-resistant (MDR)-TBM based on microbiologic testing.

### Variation in CSF concentrations of anti-TB drugs

We first sought to characterize the interindividual variability of anti-TB drug concentrations in the serum and CSF. As per standard of care for patients hospitalized with TBM at NCTLD, lumbar punctures were performed at diagnosis (baseline) and approximately 7, 14, and 28 days after TBM treatment initiation and monthly for as long as patients were hospitalized to follow CSF cell and protein counts in response to treatment [5,14]. Because CSF samples at diagnosis were obtained when most participants had received few if any doses of antibiotics, only samples obtained ≥ 7 days after treatment start were included in this analysis. Antibiotic concentrations were quantified in the serum and CSF at each time point either 2 or 6 hours after the most recent antibiotic dose as previously described [5]. A 2- to 4-fold difference in drug concentrations was found between participants in both serum (Fig 1) and CSF (Fig 2) at 2 hours and 6 hours after the dosing of anti-TB antibiotics. This was observed across antibiotics, indicating a large amount of interindividual variability in serum and CSF drug concentrations regardless of the drug used. The median and range of antibiotic concentrations in serum and CSF at each time point are summarized in S1 Table.

**Table 1. Clinical and Demographic Characteristics of Study Participants.**

|  | TB meningitis (n = 17) |
|---|---|
| Age, median (IQR) | 36 (31–44) |
| Female sex, n (%) | 9 (53) |
| HIV, n (%) | 1 (6) |
| Positive Hepatitis C Antibody, n (%) | 2 (12) |
| TB meningitis diagnostic category, n (%) | 5 (29) |
| Definite (microbiologically confirmed) |  |
| Probable | 3 (18) |
| Possible | 9 (53) |
| CSF total WBC, median (IQR) | 209 (130–286) |
| % Lymphocytes, median (IQR) | 94 (85–96) |
| % Neutrophils, median (IQR) | 5 (2–8) |
| TB meningitis grade, n (%) |  |
| Grade 1 | 8 (47) |
| Grade 2 | 9 (53) |
| Grade 3 | 0 (0) |
| Dose of each anti-TB drug |  |
| Cycloserine, | 750 mg daily |
| Ethambutol | 1200 mg daily |
| Imipenem | 2000 mg daily[a] |
| Isoniazid | 300 mg daily[b] |
| Levofloxacin | 750 mg daily[c] |
| Linezolid | 600 mg daily |
| Moxifloxacin | 400 mg daily |
| Pyrazinamide | 1600 mg daily[d] |
| Rifampin | 600 mg daily |

[a]One participant received a daily imipenem dose of 1000 mg.

[b]One participant received a daily isoniazid dose of 600 mg.

[c]One participant received a daily levofloxacin dose of 1000 mg.

[d]One participant received a daily pyrazinamide dose of 1200 mg.

There are multiple sources of potential interindividual variability in CSF drug concentrations among patients with TBM including integrity of the blood-brain and blood-CSF barriers, variability in protein binding, and differential expression of drug transporters [26]. It is also possible that the large differences in serum concentration from person to person could explain some or all of the variability in CSF drug concentrations [27]. We first sought to determine the impact of measured clinical and demographic factors on serum drug concentrations using a mixed effects linear model to account for repeated measures of drug concentrations over the course of the study. We found that for most antibiotics measured there was no significant association between serum drug concentrations and age, sex, weight, creatinine clearance or sampling time (S2 Table). The only exceptions were cycloserine and pyrazinamide, for which there was a significant negative association between serum drug concentrations and age. Serum concentrations of cycloserine were also positively associated with male sex while serum concentrations of pyrazinamide were negatively associated with creatinine clearance.

We then sought to determine how well these factors, as well as concomitantly collected serum drug concentrations, explained the variation in CSF drug concentrations. For most but not all drugs, serum drug concentrations were significantly associated with CSF drug

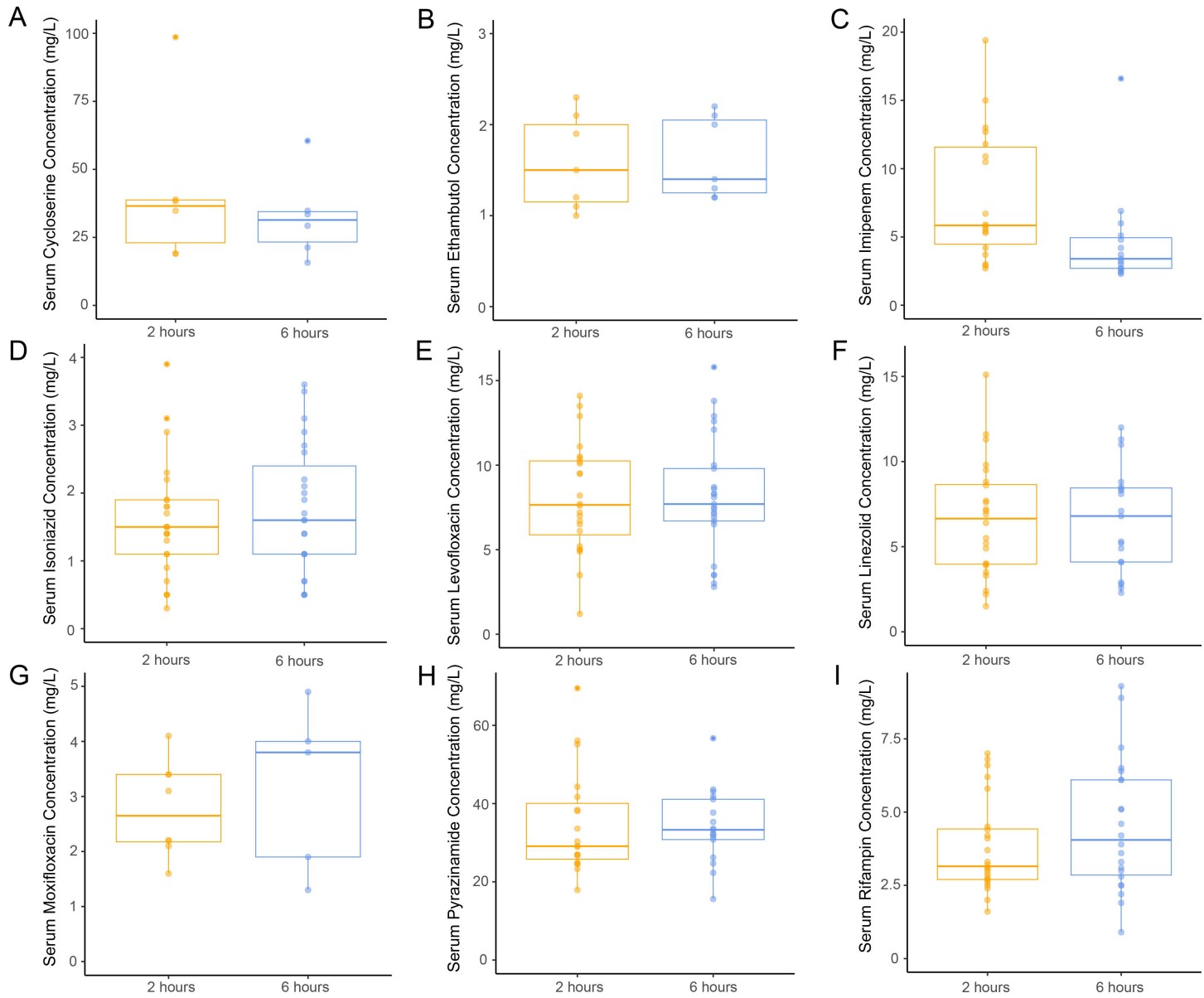

**Fig 1. Distribution of serum drug concentrations.** Boxplots depicting serum concentrations of anti-TB antibiotics 2 and 6 hours after the most recent antibiotic dose for the study participants receiving each drug: (A) cycloserine [n = 4 participants; 12 measurements], (B) ethambutol [n = 5 participants; 15 measurements], (C) imipenem [n = 10 participants; 33 measurements], (D) isoniazid [n = 14 participants; 48 measurements], (E) levofloxacin [n = 16 participants; 49 measurements], (F) linezolid [n = 11 participants; 43 measurements], (G) moxifloxacin [n = 9 participants; 13 measurements], (H) pyrazinamide [n = 11 participants; 36 measurements], and (I) rifampin [n = 13 participants; 46 measurements].

concentrations. The exceptions were imipenem and ethambutol, for which there was no significant association between serum and CSF drug concentrations (S3 Table). For cycloserine, isoniazid, linezolid, and pyrazinamide we calculated an adjusted $R^2$ of ≥0.3, suggesting serum drug concentrations explained over 30% of the variability in CSF drug concentrations (Fig 3). While the association between serum and CSF concentrations of moxifloxacin was not statistically significant (p = 0.06), this may have driven by a smaller number of samples analyzed from participants receiving this drug as serum drug concentrations explained a similar

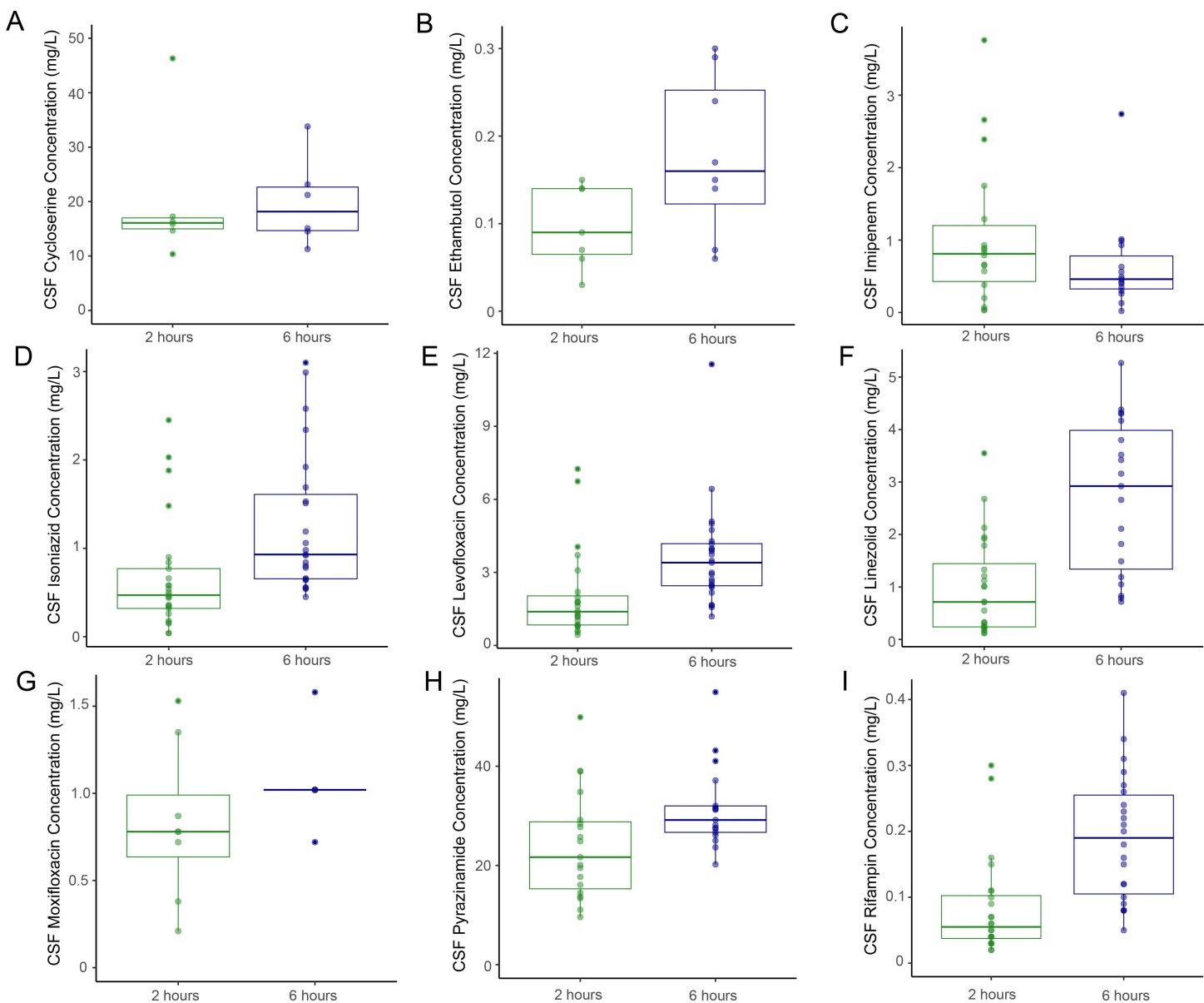

**Fig 2. Distribution of CSF drug concentrations.** Boxplots depicting cerebrospinal fluid concentrations of anti-TB antibiotics 2 and 6 hours after the most recent antibiotic dose for the study participants receiving each drug: (A) cycloserine [n = 4 participants; 12 measurements], (B) ethambutol [n = 5 participants; 15 measurements], (C) imipenem [n = 10 participants; 33 measurements], (D) isoniazid [n = 14 participants; 48 measurements], (E) levofloxacin [n = 16 participants; 49 measurements], (F) linezolid [n = 11 participants; 43 measurements], (G) moxifloxacin [n = 9 participants; 13 measurements], (H) pyrazinamide [n = 11 participants; 36 measurements], and (I) rifampin [n = 13 participants; 46 measurements].

amount of variation in CSF concentrations (a$R^2$ = 0.33). For levofloxacin and rifampin, serum drug concentrations were significantly associated with CSF concentrations, but the adjusted $R^2$ values were 0.21 and 0.16 respectively, suggesting a weaker association. Unlike serum, most CSF drug concentrations were significantly higher when sampled at the 6-hour time point with the exceptions of cycloserine and imipenem. Male sex was also associated with significantly higher CSF concentrations of levofloxacin while increased age was associated with higher concentration of ethambutol.

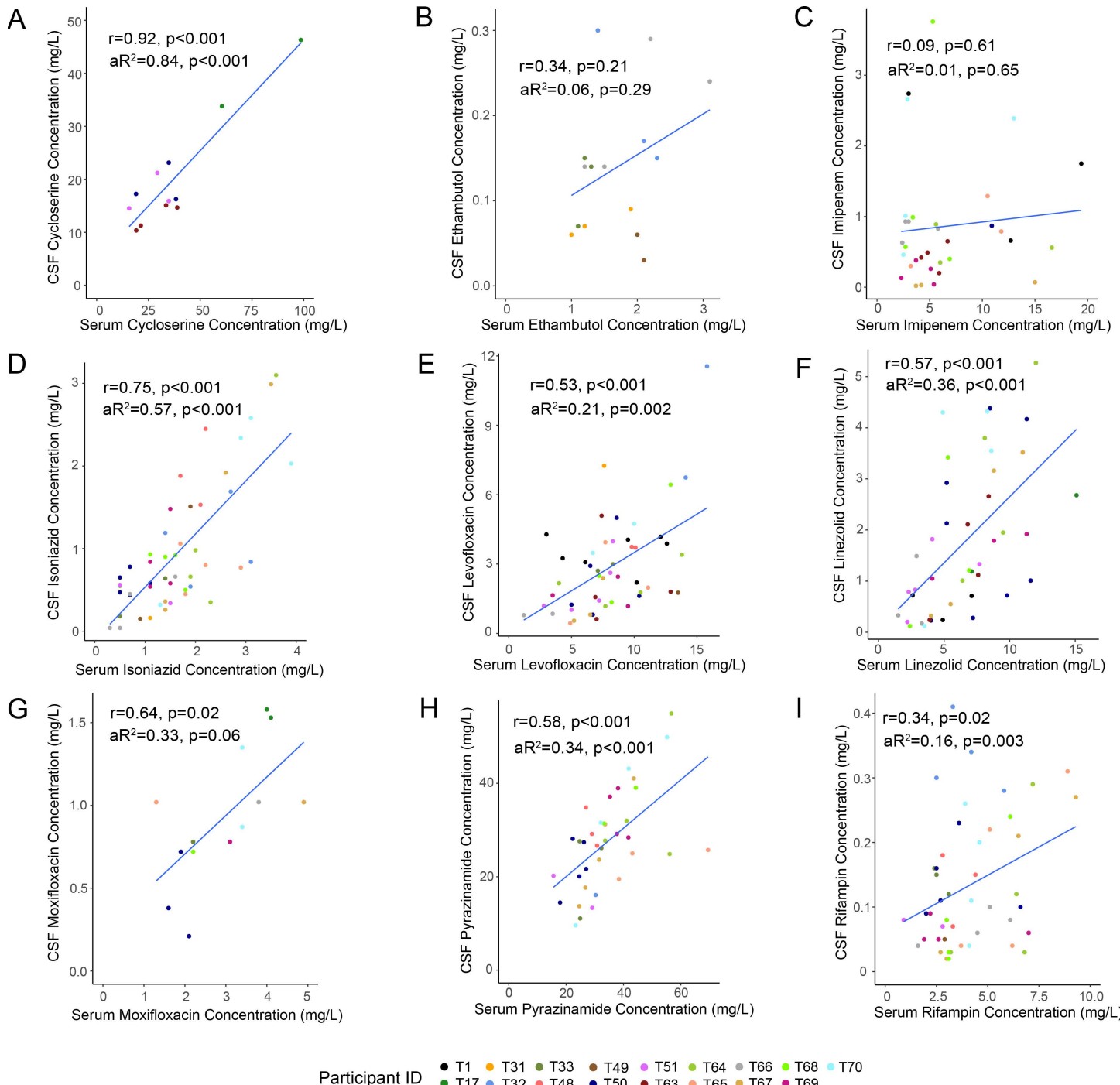

**Fig 3. Relationship between serum and CSF drug concentrations.** Scatter plots with fitted regression lines depicting the relationship between serum and cerebrospinal fluid concentrations of anti-TB drugs: (A) cycloserine, (B) ethambutol, (C) imipenem, (D) isoniazid, (E) levofloxacin, (F) linezolid, (G) moxifloxacin, (H) pyrazinamide, and (I) rifampin. For each figure, r represents the correlation coefficient with corresponding p-value. The adjusted $R^2$ was calculated using a mixed effects linear model, with p-values derived from the pairwise comparison of models including and excluding the serum concentration of each drug.

## Using untargeted metabolomics to measure CSF drug concentrations

We performed untargeted high-resolution metabolomics on CSF samples using combined liquid chromatography-mass spectrometry in dual ionization mode with HILIC positive and c18 negative chromatography. We detected 6,596 metabolic features in positive ionization mode and 9,427 in negative ionization mode. We then performed a targeted search for molecular features that were within a mass and retention time error range of 5 ppm and 30 seconds respectively for the M + H or M−H adducts of anti-TB drugs. This query yielded high confidence matches for several anti-TB drugs including ethambutol, isoniazid, pyrazinamide, rifampin, linezolid, imipenem, levofloxacin, and moxifloxacin. We then performed a correlation analysis of the peak intensity for each annotated metabolite and the antibiotic concentration as measured using MS/MS and a purified standard curve [5]. We found that the intensity value for each antibiotic as measured by our untargeted metabolomics platform was strongly correlated with the absolute concentration of most anti-TB antibiotics in the CSF. Peaks for ethambutol, pyrazinamide, isoniazid, linezolid, and imipenem all demonstrated a Pearson correlation coefficient of > 0.7 when compared with absolute drug concentrations (Fig 4; p < 0.001 for all). The peak intensities for moxifloxacin (r = 0.53, p = 0.05) and levofloxacin (r = 0.45, p < 0.001), as well as rifampin (r = 0.48, p < 0.001), were also significantly correlated with measured CSF concentrations, but the relationship was less strong than for the other antibiotics studied. In the case of rifampin, this may have been due in part to the relatively low concentration of this antibiotic in the CSF. These data indicate that untargeted metabolomics can be an effective method to approximate many anti-TB drug concentrations in biofluids.

## Relationship between CSF metabolism, inflammation, and drug concentrations

Antibiotic penetration into the CSF is generally thought to be increased when there is a breakdown in the blood-brain barrier due to meningeal inflammation in persons with TBM [28]. We therefore sought to mine concomitantly collected high-resolution metabolomics and cytokine data to determine whether any soluble immune mediators were associated with CSF drug concentrations. We quantified 32 cytokines in the CSF of the TBM participants while we were able to quantify a targeted list of 176 metabolites with confirmed chemical identities from the 16,023 total metabolic features detected [29]. We used mixed effects linear models to determine which cytokines and metabolites with known chemical identities were most strongly associated with CSF concentrations of each antibiotic. In Fig 5, we show the metabolites and cytokines significantly associated with the concentration of at least one antibiotic using a false discovery correction of q < 0.05. We found that carnitines were significantly associated with CSF concentrations of ethambutol and imipenem at q < 0.05, as well as isoniazid, linezolid, pyrazinamide, and rifampin at an unadjusted p < 0.05. However, the directionality of the association was inconsistent. While increased CSF concentrations of carnitines were associated with increased concentrations of ethambutol and imipenem, they were associated with lower CSF concentrations of isoniazid, linezolid, pyrazinamide, and rifampin. This suggests that different metabolic states in the CNS may be associated with increased drug penetration for certain classes of antibiotics while decreasing CSF drug concentrations for others.

There were few significant associations between cytokine and antibiotic concentrations in the CNS. Increased concentrations of pro-inflammatory cytokines and chemokines including TNF-a, MCP-3, IL-6, IL-18, and IFN-y were associated with increased concentrations of imipenem, but otherwise had little relationship to CSF antibiotic concentrations. The only exception was IL-6, which was significantly associated with CSF ethambutol concentrations. These data indicate that an inflammatory CSF milieu was not associated with higher CSF concentrations of most anti-TB antibiotics.

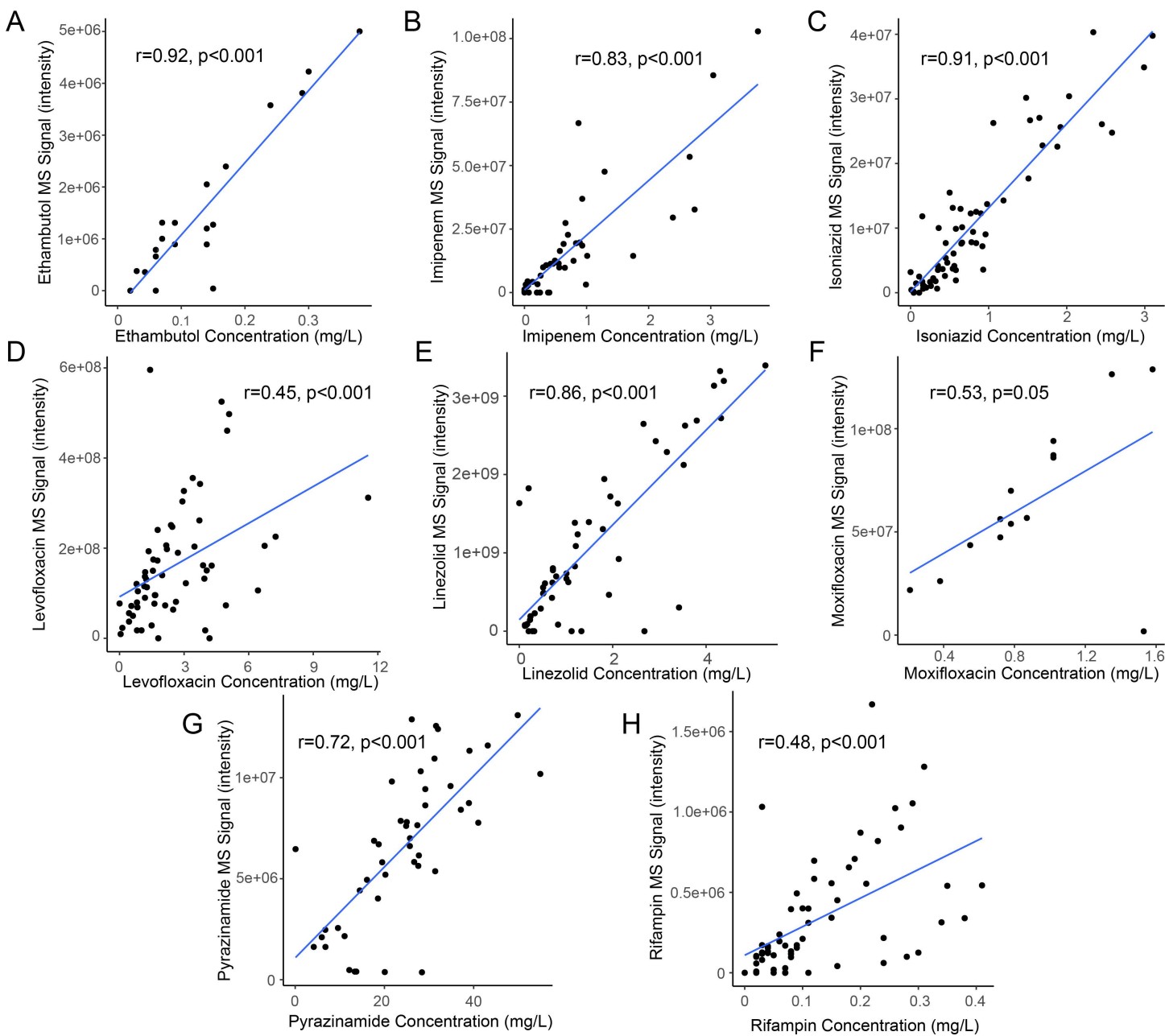

**Fig 4. Detection of TB drugs using untargeted metabolomics.** Scatter plots with fitted regression lines depicting the relationship between cerebrospinal fluid concentrations of anti-TB drugs and their mass spectrometry intensity value as measured on the untargeted metabolomics platform: (A) ethambutol, (B) imipenem, (C) isoniazid, (D) levofloxacin, (E) linezolid, (F) moxifloxicin, (G) pyrazinamide, and (H) rifampin. For each figure, r and p values are based on the Pearson correlation coefficient for each drug.

## Discussion

In this TBM pharmacometabolomics study, we show there is significant interindividual variability in CSF concentrations of anti-TB antibiotics. We further show that differences in serum drug concentrations account for much of the variability in CSF concentrations for certain drugs including linezolid, isoniazid, pyrazinamide, and cycloserine, but have a less

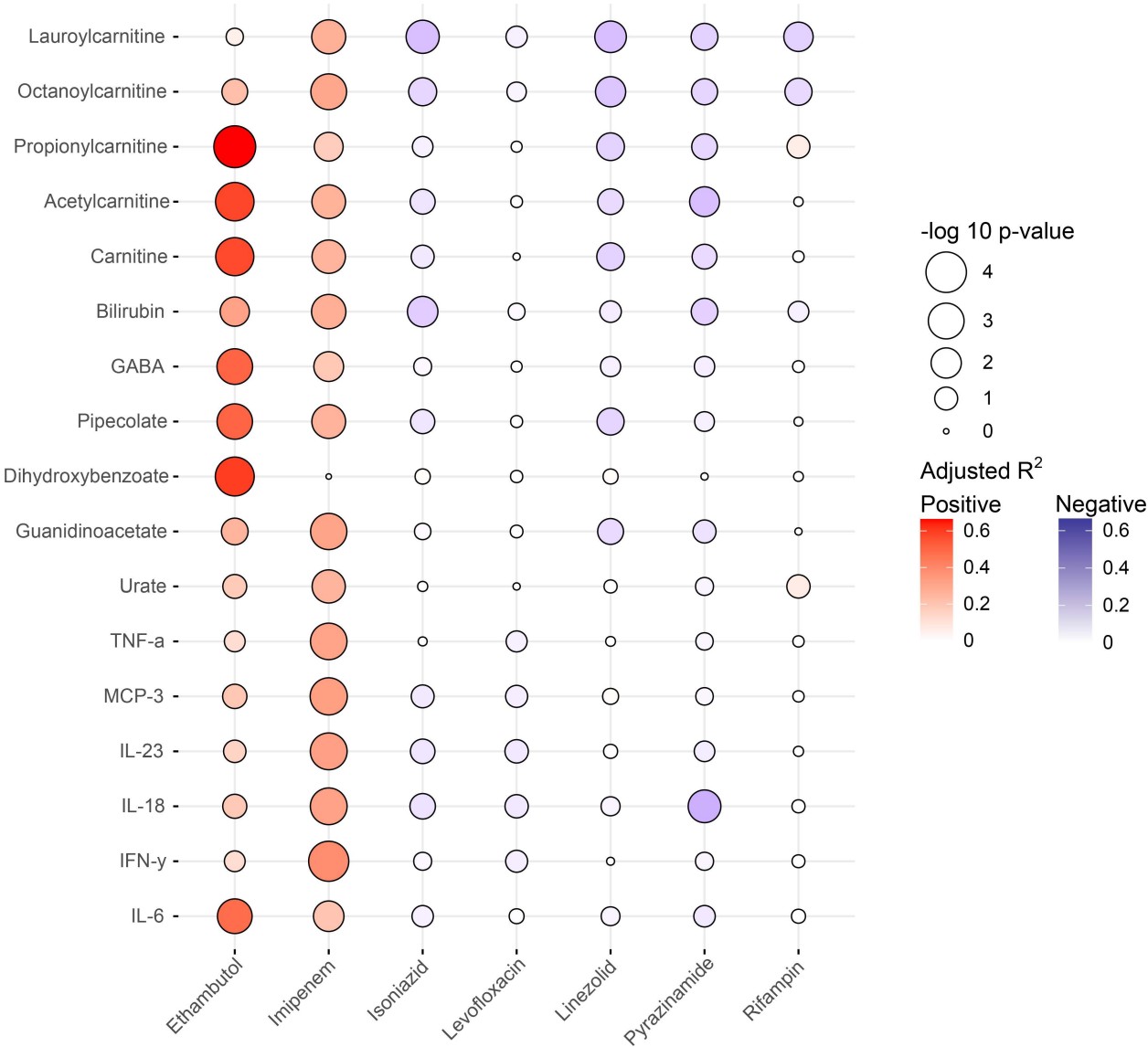

**Fig 5. Association between CSF concentrations of TB drugs and concentrations of metabolites and cytokines.** The bubble plots represent the adjusted R² values measuring the association between anti-TB drug concentrations and soluble immune mediators (metabolites, cytokines, and chemokines) in the cerebrospinal fluid. The color scale for each dot represents the adjusted R² value and dot size represents the -log P value for each association. The red scale indicates a positive association while the blue scale indicates a negative association.

strong relationship with CSF concentrations of levofloxacin and rifampin and little to no association with concentrations of imipenem and ethambutol. Finally, we demonstrate that CSF drug concentrations may be affected by the CSF metabolic milieu and have little association with concentrations of proinflammatory cytokines and chemokines. Together, these data show that a complex mix of factors contribute to CSF concentrations of anti-TB antibiotics. Factors such as meningeal inflammation and serum antibiotic concentrations are traditionally thought to be primary drivers of drug concentrations in the CNS. While this small study had a sample size of only 17 patients, we found serum drug concentrations and CSF cytokine concentrations were not associated or weakly associated with CSF drug concentrations for many antibiotics.

While serum concentrations of anti-TB drugs are known to vary widely from person to person [27], we show equally wide variability in CSF drug concentrations in those treated for TBM. Across the study population we found greater than 2-fold differences in CSF concentrations of all anti-TB drugs studied. This was true both two and six hours after the dose of each drug. Such variability in the CSF concentrations of anti-TB drugs has been shown in other studies, most notably among 237 patients (>700 CSF samples) included in a pharmacology sub study evaluating rifampin, isoniazid, and levofloxacin CSF concentrations [13]. The substantial variation in CSF concentrations among study participants was evidenced by large 95% confidence intervals for both $C_{max}$ and AUC parameters [13]. Furthermore, interpatient variability in CSF rifampin [12] and fluoroquinolone [30] concentrations have been demonstrated in clinical trial dose finding and drug comparison studies, respectively. Penetration across the blood-CSF barrier depends on the chemical structure of the drug as well as additional factors including the condition of the blood/CNS barrier and additional drugs in the treatment regimen. The structure, molecular size, lipophilicity, and ionization status at a pH of 7.4 for each drug also impact their ability to cross the blood-CSF barrier. What has been less studied, is clinical or laboratory predictors of CSF drug concentrations. We found that serum concentrations are poorly predictive of CSF concentrations for many drugs including imipenem, rifampin, and ethambutol. Using the plasma rifampin $AUC_{0-24}$ parameter, Dian et al, found a high correlation between plasma and CSF rifampin concentrations (Spearman's $\rho$ 0.7, $p < 0.01$) [12]. This stronger relationship could have been due to the fact that $AUC_{0-24}$ values rather than point-in-time concentrations were used in the analysis, which better account for differences in CSF drug penetration over time. A better understanding of predictors of CSF drug penetration is needed to help both understand the underlying mechanisms of drugs reaching the site of disease and to identify clinically useful markers to help guide optimal drug dosing in TBM.

This study provides evidence that untargeted, high-resolution metabolomics can be used to approximate drug concentrations of multiple anti-TB drugs in biofluids including imipenem, ethambutol, isoniazid, linezolid, and pyrazinamide. For other drugs such as moxifloxacin, levofloxacin, and rifampin, peak intensities were more weakly associated with absolute concentrations. For these drugs, caution should be exercised when quantifying drug concentrations from untargeted data, particularly in instances of low concentration. Our results are supported by studies showing untargeted metabolomics can also be used to capture exposure to xenobiotics in the environment [8]. The ability to approximate drug concentrations as part of an untargeted metabolomics platform is likely to create additional opportunities to understand drug concentrations and pharmacokinetics in the context of broader measurement of host metabolism. Understanding broad host metabolic responses to drugs, termed "pharmacometabolomics", is an emerging area of study with great potential to improve our understanding of heterogenous host responses to medications including antibiotics [11]. However, a limitation of the approach has been the need to separately perform targeted quantification of drugs and untargeted metabolomics. Our findings suggest that in many cases, important insights in pharmacometabolomics could be obtained from untargeted metabolomics data alone.

By simultaneously collecting pharmacology, metabolomics, and cytokine data in the present study, we were able to examine which soluble immune and metabolic mediators in CSF are associated with improved CNS penetration of TB drugs. The association between CSF concentrations of multiple antibiotics and carnitines suggests that a metabolic milieu with impaired oxidative phosphorylation may impact antibiotic penetration in the CNS. Increased carnitines in the CSF, which transport fatty acids from the cytosol into mitochondria to be oxidized, may reflect decreased transport of free fatty acids into mitochondria due to elevated levels of glycolysis, which tends to produce greater inflammatory signaling in immune cells [9,31,32]. Historically, greater inflammation in the CNS has been thought to be a catalyst for

drug penetration across blood-brain and blood-CSF barriers [28]. The present study found little direct relationship between CSF drug concentrations and pro-inflammatory cytokines. While elevated CSF concentrations of cytokines does not necessarily indicate increased meningeal inflammation, the findings do suggest that the relationship between CSF drug concentrations and inflammation may be weak. This is supported by prior work from our group, which indicated CSF concentrations of anti-TB drugs are roughly equal early in the course TBM treatment versus two or more months after treatment start, when meningeal inflammation would be expected to decrease [5]. Another explanation for these findings is that the CSF milieu remains highly inflammatory months after treatment start [7], which could mean CNS drug penetration is enhanced for an extended period of time. However, it is also important to note that the present study included no cases of grade 3 TBM, so it remains possible that those with the greatest CSF inflammatory response do experience increased drug concentrations.

This study is subject to several limitations. Due to the small sample size of persons with TBM and low number of microbiologically confirmed cases, this study had limited statistical power to detect factors associated with CSF drug concentrations. Additionally, the study was conducted in a single geographic region, potentially limiting generalizability. CSF albumin and the protein binding of anti-TB drugs were not measured, limiting our ability to evaluate the impact of these factors on the observed variation in CSF drug concentrations. Though we demonstrate a strong association between CSF concentrations of multiple metabolites and concentrations of anti-TB drugs, the observational nature of the study precludes us from establishing a causal relationship. In future studies it will be important to evaluate the links between key metabolites and CSF drug penetration using animal models of TBM and larger, diverse human cohorts. Enhancing our understanding of how soluble mediators in the CNS impact drug penetration may lead to therapies that enhance drug delivery to this area or personalized dosing regimens for individual patients.

Overall, this study shows the interindividual variation in CSF drug concentrations among persons with TBM is high and potentially linked to the metabolic milieu of the CSF. Serum drug concentrations are only weakly associated with CSF concentrations for some drugs, and the amount of inflammation in the CSF appears to have a similarly weak association with CSF drug penetration. Our results provide insight into the factors that impact CSF drug concentrations in TBM and indicate that improved understanding of the host metabolic response could provide targets to enhance drug delivery.

## Supporting information

**S1 Table. Serum and CSF drug concentrations 2 and 6 hours after most recent antibiotic dose.**
(DOCX)

**S2 Table. Association between clinical and demographic factors and serum drug concentrations.**
(DOCX)

**S3 Table. Association between clinical and demographic factors and CSF drug concentrations.**
(DOCX)

**S1 File. Study datasets.** The attached datasets include the CSF and serum drug concentrations, clinical and demographic information, and CSF metabolite and cytokine concentrations for each study participant.
(XLSX)

## Acknowledgments

The authors thank the physicians, nurses, and staff at the NCTLD in Tbilisi, Georgia, who provided care for the patients with TBM included in this study. Additionally, the authors are thankful for study participants with tuberculosis meningitis who were willing to participate in the study and help contribute meaningful data that may help future patients with the same illness.

## Author contributions

**Conceptualization:** Jeffrey M. Collins, Maia Kipiani, Yutong Jin, Henry M. Blumberg, Charles Peloquin, Russell R. Kempker.

**Data curation:** Jeffrey M. Collins, Maia Kipiani, Ashish A. Sharma, David Benkeser.

**Formal analysis:** Jeffrey M. Collins, Yutong Jin, Jeffrey A. Tomalka, David Benkeser.

**Funding acquisition:** Maia Kipiani, Russell R. Kempker.

**Investigation:** Maia Kipiani, Ashish A. Sharma, Jeffrey A. Tomalka, Teona Avaliani, Mariam Gujabidze, Tinatin Bakuradze, Shorena Sabanadze, Zaza Avaliani, Henry M. Blumberg, Dean P. Jones, Charles Peloquin, Russell R. Kempker.

**Methodology:** Jeffrey M. Collins, Ashish A. Sharma, Jeffrey A. Tomalka, Teona Avaliani, Shorena Sabanadze, Zaza Avaliani, David Benkeser, Dean P. Jones, Charles Peloquin, Russell R. Kempker.

**Project administration:** Maia Kipiani, Teona Avaliani, Mariam Gujabidze, Tinatin Bakuradze, Shorena Sabanadze, Zaza Avaliani, Russell R. Kempker.

**Resources:** Maia Kipiani, Jeffrey A. Tomalka, Mariam Gujabidze, Tinatin Bakuradze, Shorena Sabanadze, Zaza Avaliani, Dean P. Jones, Charles Peloquin.

**Software:** Jeffrey M. Collins, Yutong Jin, Dean P Jones.

**Supervision:** Maia Kipiani, Zaza Avaliani, Henry M. Blumberg, David Benkeser, Dean P. Jones, Russell R. Kempker.

**Validation:** Jeffrey M. Collins, Dean P. Jones, Charles Peloquin.

**Visualization:** Jeffrey M. Collins, Yutong Jin, David Benkeser.

**Writing – original draft:** Jeffrey M. Collins.

**Writing – review & editing:** Jeffrey M. Collins, Maia Kipiani, Yutong Jin, Ashish A Sharma, Jeffrey A. Tomalka, Teona Avaliani, Mariam Gujabidze, Tinatin Bakuradze, Shorena Sabanadze, Zaza Avaliani, Henry M Blumberg, David Benkeser, Dean P Jones, Charles Peloquin, Russell R. Kempker.

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
