## [Decision Letter · Decision Letter 0]

8 Oct 2024

PONE-D-24-31354Pharmacometabolomics in TB Meningitis – understanding the pharmacokinetic, metabolic, and immune factors associated with anti-TB drug concentrations in cerebrospinal fluidPLOS ONE

Dear Dr. Collins,

Thank you for submitting your manuscript to PLOS ONE. After careful consideration, we feel that it has merit but does not fully meet PLOS ONE’s publication criteria as it currently stands. Therefore, we invite you to submit a revised version of the manuscript that addresses the points raised during the review process.

We look forward to receiving your revised manuscript.

Kind regards,

Selvakumar Subbian, Ph.D.

Academic Editor

PLOS ONE

Journal Requirements:

2. Thank you for stating the following financial disclosure: This work was supported by grants from the National Institutes of Health (NIH) and National Institute of Allergy and Infectious Diseases [R03AI139871, K23AI103044, K23AI144040, P30AI168386, P30AI050409]; NIH Fogarty International Center [D43TW007124]; and NIH National Center for Advancing Translational Science [UL1TR002378], Bethesda, MD, USA.  

Reviewers' comments:

Reviewer's Responses to Questions

**Comments to the Author**

1. Is the manuscript technically sound, and do the data support the conclusions?

Reviewer #1: Yes

Reviewer #2: Yes

2. Has the statistical analysis been performed appropriately and rigorously? 

Reviewer #1: Yes

Reviewer #2: Yes

3. Have the authors made all data underlying the findings in their manuscript fully available?

Reviewer #1: Yes

Reviewer #2: Yes

4. Is the manuscript presented in an intelligible fashion and written in standard English?

Reviewer #1: Yes

Reviewer #2: Yes

5. Review Comments to the Author

Reviewer #1: This study attempts to understand mechanisms to the variability of CSF concentrations of anti-TB drugs in a devastating disease, TB meningitis. This is a crucial translational research question that should be examined to help optimize drug therapy strategies for TBM. A few comments for consideration:

Abstract

- Conclusion of abstract (line 21-22) – it says CSF conc were not associated with cytokines in the CSF but the sentence right before says imipenem CSF conc were associated with CSF cytokine conc – this inconsistency needs to be reconciled.

Methods

- Stats analysis – line 122-125 – to confirm, ref 22 demonstrates the use of the R package, mitmil and ref 23 demonstrates this statistical approach? Are the refs correct here?

Results

- Participants – any data on comorbidities? Was this collected? I don’t expected much diabetes in such a young cohort, but any patients with HIV or any other notable comorbidities?

- Line 153-155 – is the reason to use later samples (>7 days) in order to taper off any differences that would be at baseline in terms of steady state vs non steady state conc? If so, that clarification would be of interest to readers as to the decision of not using baseline CSF samples.

- Line 164 and 166 – are refs 26 and 27 correct? They look like papers on LCMS and untargeted metabolomics – double check if this is the appropriate citation.

- Line 187 – the point about moxi having a smaller number of participants taking it is a good point. I don’t see a listing of the n for each antibiotic throughout the manuscript. This should be laid out – either in a small table (in the supplement?) or in table 1 or in the caption for the figures. It should be clear the n for each antibiotic, each specimen type (serum, CSF), and number of samples included in the summary plot since this may differ across drugs.

- General comment – is there data on dosing? What are the median doses for each antibiotic? Can that be added to table 1?

Discussion

- In lines 252-253, it discusses how serum drug conc account for variability in CSF conc but then in lines 260-263 states that serum drug conc are not/weakly associated with CSF drug conc. And in lines 180-181 it states serum drug conc are significantly associated with CSF. This all appears to be inconsistent and needs to be clarified.

- End of page 15/start of page 16 – discussion on inflammation. All these patients received dexamethasone (as is standard of care) – how does that factor into this discussion on inflammatory state and drug penetration into CSF? How does this (based on hypothesis-generating theory or other prior literature) impact the relationship between serum and CSF drug concentrations?

- Part of the limitations of this study is the drug conc sampling – is there a distributional delay from serum into CSF which could explain the weak association for some drugs or is this a non issue given the half life of the drugs and likely being at steady state? Something to comment on if you feel its notable.

References

- Refs 29-32 seem as they should correspond to the Methods (stats analysis) section but are rather used in seemingly unrelated (maybe erroneous places) – e.g., line 274 – ref 30 is tagged to ‘fluoroquinolone’ in regard to CSF conc but this is a ref on the nlme R package. Or is ref 28 appropriate for line 222? Ref 27 at line 265? Please check references throughout to ensure there is no systemic error.

Reviewer #2: In this study, the authors have reported the variability in antibiotics presence in serum and CSF in TBM patients. The study raises a significant question towards the effective treatment of TBM. I have following concerns regarding the manuscript,

1. The variability in drug concentration in body fluids is already known. The authors need to explain the added advantage of their study.

2. Did they study the disease outcome and particular drug concentration in serum and CSF? This is significant question and may give an idea for choice of drug for TBM patients.

3. Please add plausible explanation for variability in drug concentration. Is there any relation with the chemical structure of the drug?

6. PLOS authors have the option to publish the peer review history of their article (what does this mean? ). If published, this will include your full peer review and any attached files.

**Do you want your identity to be public for this peer review?** For information about this choice, including consent withdrawal, please see our Privacy Policy .

Reviewer #1: No

Reviewer #2: No

---

## [Author Response · Author response to Decision Letter 0]

8 Nov 2024

Journal Requirements

1. Comment: Please state what role the funders took in the study.

Response: We confirm the following, “The funders had no role in study design, data collection and analysis, decision to publish, or preparation of the manuscript.” We have also included this statement in the Cover Letter. Additionally, we included a similar statement at the end of the manuscript [page 22, lines 454-455].

2. Comment: Please include your full ethics statement in the Methods section of your manuscript file.

Response: We have included the full name of the IRBs who approved of our study and that informed written consent was obtained from all participants as follows [page 5, line 76], “Written informed consent was obtained from all study participants, and study approval was obtained from the institutional review boards of Emory University and the NCTLD.”

Reviewer 1

3. Comment: In the conclusion of the abstract (line 21-22) – it stays CSF conc were not associated with cytokines in the CSF but the sentence right before says imipenem CSF conc were associated with CSF cytokine conc- this inconsistency needs to be reconciled.

Response: Thank you for this comment. The reviewer correctly points out that there were certain cytokines associated with CSF concentrations of imipenem. We have therefore edited the abstract conclusion statement to read “With the exception of imipenem, there was no association between CSF drug concentrations and concentrations of cytokines and chemokines” [page 2, lines 21-23].

4. Comment: In the methods under Stats analysis – line 122-125 – to confirm, ref 22 demonstrates the use of the R package, mitmil and ref 23 demonstrates this statistical approach? Are the refs correct here?

Response: The reviewer correctly points out that these references were not properly numbered. This has been corrected in the revised bibliography.

5. Comment: Participants – any data on comorbidities? Was this collected? I don’t expect much diabetes in such a young cohort, but any patients with HIV or any other notable comorbidities?

Response: Thank you for this comment. We collected information on comorbidities and have included it in Table 1 as suggested. We included the number of participants with HIV and Chronic Hepatitis C; both had a low prevalence among our small cohort. No participant had diabetes.

6. Comment: Line 153-155 – Is the reason to use later samples (>7 days) to taper off any differences that would be at baseline in terms of steady state vs non steady state conc? If so, that clarification would be of interest to readers as to the decision of not to use baseline CSF samples.

Response: We did not test baseline samples for drug concentrations as most of the participants were not receiving antibiotics at the time of initial lumbar puncture. We clarified this point in the methods [page 10, lines 169-170].

7. Comment: Line 164 and 166 – are refs 26 and 27 correct? They look like papers on LCMS and untargeted metabolomics – double check if this is the appropriate citation.

Response: Similar to above, the reviewer correctly points out that these references were not properly numbered. This has been corrected in the revised bibliography.

8. Comments: Line 187 – the point about moxi having a smaller number of participants taking it is a good point. I don’t see a listing of the n for each antibiotic throughout the manuscript. This should be laid out – either in a small table (in the supplement?) or in table 1 or in the caption for the figures. It should be clear the n for each antibiotic, each specimen type (serum, CSF), and number of samples included in the summary plot since this may differ across drugs.

Response: We have added the number of total participants and samples analyzed for each drug in the legends of Figure 1 and Figure 2. We also clarified that while several participants had CSF samples analyzed for moxifloxacin concentration, there was a small number of total CSF samples analyzed.

9. Comment: General comment – is there data on dosing? What are the median doses for each antibiotic? Can that be added to table 1?

Response: With only a small number of exceptions, each participant taking a particular anti-TB drug received the same dose. The dose of each drug used has been added to Table 1. For four drugs, there was one participant that received a different dose. These exceptions were added as footnotes to the table.

10. Comment: In lines 252-253, it discusses how serum drug conc account for variability in CSF conc but then in lines 260-263 states that serum drug conc are not/weakly associated with CSF drug conc. And in lines 180-181 it states serum drug conc are significantly associated with CSF. This all appears to be inconsistent and needs to be clarified.

Response: There were significant associations between the serum and CSF drug concentrations for most but not all anti-TB drugs and we have tried to make the wording clearer on this point. We amended Line 182 [page 12, line 222 in the revised manuscript] as follows, “For most but not all drugs, serum drug concentrations were significantly associated with CSF drug concentrations.” and Line 253 [page 17, line 336 in the revised manuscript], “We further show that differences in serum drug concentrations account for much of the variability in CSF concentrations for certain drugs including...” to make this clearer. Each sentence is then followed by further details about the associations between serum and CSF drug concentrations for each of the anti-TB drugs tested.

11. Comment: End of page 15/start of page 16 – discussion on inflammation. All these patients received dexamethasone (as is standard of care) – how does that factor into this discussion on inflammatory state and drug penetration into CSF? How does this (based on hypothesis-generating theory or other prior literature) impact the relationship between serum and CSF drug concentrations?

Response: Given that all patients received Dexamethasone, we were not able to measure/evaluate the impact of steroids on the host inflammatory response or CSF drug penetration. The conventional thinking is that steroids may reduce the penetration of drugs across the blood CNS barriers and while we were unable to test this hypothesis, our previous work has found that most CSF drug concentrations are stable or increased over the first few months of treatment. [PMCIDs PMC9791083, PMC9464073].

12. Comment: Part of the limitations of this study is the drug conc sampling – is there a distributional delay from serum into CSF which could explain the weak association for some drugs or is this a non-issue given the half-life of the drugs and likely being at steady state? Something to comment on if you feel it's notable.

Response: The reviewer raises an important point, and this was the reason that we alternated CSF sampling at each time point between 2 and 6 hours. We found 2- and 6-hour CSF drug concentrations did not significantly differ for cycloserine, imipenem, or moxifloxacin. However, we did find significantly higher CSF concentrations at 6 hours for ethambutol, isoniazid, linezolid, levofloxacin, pyrazinamide, and rifampin (indicating delayed CSF penetration). We controlled for the sampling time in linear mixed effects regression models looking at the association of serum and CSF concentrations to account for any differences in CSF penetration. The results of these models are shown in Supplemental Table 3.

13. Comment: - Refs 29-32 seem as they should correspond to the Methods (stats analysis) section but are rather used in seemingly unrelated (maybe erroneous places) – e.g., line 274 – ref 30 is tagged to ‘fluoroquinolone’ in regard to CSF conc but this is a ref on the nlme R package. Or is ref 28 appropriate for line 222? Ref 27 at line 265? Please check references throughout to ensure there is no systemic error.

Response: We thank the reviewer for the careful review of the references. These references were also not numbered correctly. This has been corrected in the revised bibliography.

Reviewer 2

14. Comment: The variability in drug concentration in body fluids is already known. The authors need to explain the added advantages of their study.

Response: Thank you for the comment. A major advantage of our study was that it looked at the association between soluble molecules (metabolites and cytokines) and CSF concentrations of anti-TB drugs. Given the high variability (intra patient and inter patient) of CSF anti-TB drug concentrations, it is imperative to determine if there are any biomarkers that can predict CSF drug penetration. Our study represents the most comprehensive study to date looking at the association of metabolites and cytokines and CSF-TB Drug concentrations. An additional advantage of our study is that it found that untargeted, high-resolution metabolomics could accurately identify and approximate drug concentrations of certain anti-TB drugs in the CSF including imipenem, ethambutol, isoniazid, linezolid, and pyrazinamide. This finding will open new avenues in the study of host metabolism and pharmacokinetics. We have highlighted these points in the discussion.

15. Comment: Did they study the disease outcome and particular drug concentration in serum and CSF? This is a significant question and may give an idea for the choice of drug for TBM patients.

Response: Thank you for raising this point. This study had a small sample size and only included patients with Grade 1 or 2 disease, all of whom had a favorable outcome. Because there weren’t any participants in the cohort that had a measurably poor outcome (e.g. death), it was not feasible to perform an analysis looking at the association of CSF drug concentrations and clinical outcomes. Some of the ongoing TBM clinical trials are incorporating pharmacokinetics and hopefully will be able to evaluate the association between PK/PD measures and clinical outcomes [PMCID: PMC7616680].

16. Comment: Please add plausible explanation for variability in drug concentration. Is there any relation with the chemical structure of the drug?

Response: This is a good and important point. Penetration across the blood-cerebrospinal fluid barrier does depend on the chemical structure of the drug as well as additional factors including the host inflammatory state, condition of the blood/CNS barrier, and additional drugs being received. Regarding the structure of the drug, molecular size, lipophilicity, and ionization status at a pH of 7.4 all in part determine the ability of each drug to cross the blood-CSF barrier. We have added these points to the revised discussion section [page 18, lines 360-364].

---

## [Decision Letter · Decision Letter 1]

5 Dec 2024

Pharmacometabolomics in TB Meningitis – understanding the pharmacokinetic, metabolic, and immune factors associated with anti-TB drug concentrations in cerebrospinal fluid

PONE-D-24-31354R1

Dear Dr. Collins,

We’re pleased to inform you that your manuscript has been judged scientifically suitable for publication and will be formally accepted for publication once it meets all outstanding technical requirements.

Kind regards,

Selvakumar Subbian, Ph.D.

Academic Editor

PLOS ONE
---

## [Editor Report · Acceptance letter]

PONE-D-24-31354R1

PLOS ONE

Dear Dr. Collins,

I'm pleased to inform you that your manuscript has been deemed suitable for publication in PLOS ONE. Congratulations! Your manuscript is now being handed over to our production team.

Kind regards,

on behalf of

Dr. Selvakumar Subbian

Academic Editor

PLOS ONE